# High Prevalence of *Klebsiella pneumoniae* in Greek Meat Products: Detection of Virulence and Antimicrobial Resistance Genes by Molecular Techniques

**DOI:** 10.3390/foods11050708

**Published:** 2022-02-28

**Authors:** Nikoletta Argyro Theocharidi, Iliana Balta, Dimitra Houhoula, Andreas G. Tsantes, George P. Lalliotis, Angeliki C. Polydera, Haralambos Stamatis, Panagiotis Halvatsiotis

**Affiliations:** 1Department of Food Science and Technology, Faculty of Food Sciences, University of West Attica, 12243 Athens, Greece; ft17198@uniwa.gr (N.A.T.); ft17214@uniwa.gr (I.B.); 2Microbiology Department, ‘Saint Savvas’ Oncology Hospital, 11522 Athens, Greece; andreas.tsantes@yahoo.com; 3Department of Animal Science, Agricultural University of Athens, Iera Odos 75, 11855 Athens, Greece; glaliotis@aua.gr; 4Biotechnology Laboratory, Department of Biological Applications and Technologies, University of Ioannina, 45110 Ioannina, Greece; apolydera@uoi.gr (A.C.P.); hstamati@uoi.gr (H.S.); 52nd Propaedeutic Department of Internal Medicine, Medical School, National and Kapodistrian University of Athens, “ATTIKON” University Hospital, 12461 Chaidari, Greece; pahalv@gmail.com

**Keywords:** *Klebsiella pneumoniae*, food, antimicrobial resistance, virulence

## Abstract

**Background:** The presence of antimicrobial-resistant pathogens such as *Klebsiella pneumoniae* strains in the food supply is dangerous. The aim of this study was to assess the prevalence of *Klebsiella pneumonia* strains in Greek meat products and evaluate their phenotypes and genotypes. **Methods:** One hundred and ten meat specimens were cultured for the isolation of *K. pneumoniae*. In positive specimens, PCR (Polymerase Chain Reaction) analysis was performed to confirm the presence of *K. pneumoniae*. Genotypic and phenotypic evaluation of the isolated strains included multiplex immunoassay for the detection of carbapenemases, and PCR screening for the detection of resistance and virulence genes. **Results:**
*K. pneumoniae* strains were recovered in 90 (81.8%) meat samples. The *ecpA* gene was identified in 30 (33.3%) isolates, while the *fimH-1* and *mrkA* genes were present in 15 (16.7%) and 65 (72.2%) isolates, respectively. Sixty-five *K. pneumoniae* isolates (72.2%) were found to carry at least one resistance gene; of these, the *bla**NDM*-like was the most commonly identified gene in 40 (61.5%) isolates, followed by the *blaOXA*-48 like gene in 20 isolates (30.8%). **Conclusions:** A high frequency of foodborne *K. pneumoniae* in Greece was found. Our results indicate that most strains carried resistance and virulence genes, indicating a high pathogenic potential and a significant risk to human health.

## 1. Introduction

Resistance to antibiotics is a global public health problem [1], hindering the treatment and extirpation of a growing number of bacterial, fungal and viral infections. Antibiotic-resistant bacteria are associated with high mortality in septic patients [2]. The rapid development of antibiotic-resistant strains is favored by the widespread and, in many cases, unjustified use of broad-spectrum antibiotics [3]. Resistance genes can also spread among different bacteria populations through the horizontal gene transfer (HGT) of mobile genetic elements (MGE) [4,5], facilitating the insurgence of multi-drug resistant strains [2,6,7].

*K. pneumoniae* is a common cause of community-acquired and nosocomial infections such as urinary tract infections, lower respiratory infections and liver abscesses. These pathogens can be found in various microbiological niches such as food and soil, but also in the skin, intestines and feces of mammals. Although *K. pneumoniae* it is not traditionally considered as a foodborne pathogen, there are reports of *K. pneumoniae* infections preceded by gut colonization, supporting the theory of food as a possible transmission vector for these pathogens [8,9,10,11]. *Enterobacterales* resistant to carbapenems and third-generation cephalosporin such as *Klebsiella pneumoniae* have been listed by the World Health Organization (WHO) as one of the top threats to public health worldwide, highlighting the need for the development of new treatment medications [6,7]. There are different resistance genes in *K. pneumoniae* which are located in transferable genetic elements and may be transferred to other bacteria. Unfortunately, antimicrobial-resistant *K. pneumoniae* strains have been detected in various food samples, such as raw meat, marketed fresh vegetables, ready-to-eat meals, and seafood [12,13,14,15,16,17]. Isolation of antimicrobial-resistant *K. pneumoniae* strains in the food supply is alarming, since they can become a large pool for resistance genes, carrying the risk for the emergence of new resistance mechanisms. 

The aim of this study was to assess the contamination rate of Greek meat samples from different local markets with *K. pneumoniae strains* and evaluate the phenotypic and molecular characteristics of these strains. As there is a lack of data, limited information exists about the presence of highly resistant strains such as *K. pneumoniae* in meat products that are intended for consumption in market chains in Greece.

## 2. Materials and Methods

### 2.1. General Workflow

A general workflow was designed in order to understand the prevalence of *Klebsiella pneumoniae* in meat products. This study involved molecular screening of isolated *Klebsiella pneumoniae.* The general workflow of the study is provided in Figure 1.

### 2.2. Sample Collection

A total of 110 meat samples were collected from different local markets in Athens. The samples included 40 (36.4%) meat specimens from pork, 35 (31.8%) from chicken and 35 (31.8%) from bovine. 

### 2.3. Detection of K. pneumoniae Colonies

Twenty-five grams from each sample was collected in sterile bags and 225 mL of buffered peptone water (BPW) solution 1% was added and homogenized in a Stomacher (Laboratory Blender Stomacher 400, Seward Medical, London, UK). The sample suspensions were incubated overnight at 37 °C. A 100 μL aliquot of the pre-enrichment culture was streaked directly onto selective CHROMagar ORIENTATION (Bioprepare Microbiology, Athens, Greece) and incubated for 24 h at 37 °C. The specimens were inoculated and incubated on MacConkey agar. After 24–48 h of aerobic incubation at 36–37 °C, isolates with colonial appearance of *Escherichia coli* and *Klebsiella pneumoniae* characterized according to Gram staining reaction and motility testing method. Suspected isolates of *Escherichia coli* and *Klebsiella* species were confirmed. Colonies exhibiting red and metallic blue color from CHROMagar ORIENTATION were picked and regrown on tryptic soy agar (TSA; Oxoid) (Bioprepare Microbiology, Athens, Greece) for 24 h at 37 °C in order to have fresh cultures needed for further DNA isolation and confirmation by PCR. 

### 2.4. DNA Isolation

Nucleic acids were extracted from the isolated colonies using the Zybio Nucleic Acids Isolation kit (Magnetic Beads Method) with an automatic extractor Zybio EXM 3000, following the manufacturer’s instructions. The DNA samples were separated into six aliquots for further analyses. 

### 2.5. Confirmation of E. coli and K. pneumoniae by PCR

PCR assays were performed using GoTaq Hot Start Master Mix (Promega Gmbh, 68199, Mannheim, Germany), 0.2 mM of each gene’s specific primers (Table 1) [16] and 10 μL of eluted DNA and ddH_2_O (molecular grade) to bring the volume to 50 μL. PCR reactions were conducted in a thermal cycler (Applied Biosystems Athens, Greece). Amplification conditions consisted of an initial 5 min denaturation step at 95 °C, followed by 40 cycles of 35 s denaturation at 94 °C, 60 s annealing at 54 °C, and a 60 s extension step at 72 °C, and a final extension step at 72 °C for 10 min. PCR products were separated in a 2% agarose gel, stained with ethidium bromide (0.5 μg/mL). The genomic DNA of a clinical strain *of Salmonella*
*enteritis* was used as a negative control. 

### 2.6. Detection of the Virulence Genes ecpA, fimH, mrkA and the Resistance Genes blaKPC, blaNDM, blaVIM, bla IMP and blaOXA-48 by Multiplex PCR

We proceeded to conduct four multiplex assays to detect three combination sets of resistant and virulence genes. Set 1 referred to *blaVIM* and *blaKPC* genes, set 2 referred to *blaNDM*, Set 3 referred to *blaIMP* and *blaOXA-48* genes and set 4 referred to *ecpA*, *fimH*, and *mrkA* genes. PCR assay were performed using GoTaq Hot Start Master Mix (Promega Gmbh, 68199, Mannheim, Germany), 0.2 mM of each gene’s specific primers (Table 1) [17,18,19], and 10 μL of eluted DNA and dH_2_O (molecular grade) to bring the volume up to 50 μL. PCR reactions were conducted in a thermal cycler (Veriti™ 96-Well Thermal Cycler, Applied Biosystems, Athens Greece). Amplification conditions consisted of an initial 5 min denaturation step at 95 °C, followed by 40 cycles of 35 s denaturation at 94 °C, 60 s annealing at 58 °C for set 4 and an annealing at 56 °C for the set 1, 2, 3, and a 60 s extension step at 72 °C, and a final extension step at 72 °C for 10 min. PCR products were separated in a 2% agarose gel, stained with ethidium bromide (0.5 μg/mL). 

### 2.7. Phenotypic Analysis

Phenotypic analysis included an in vitro multiplex immunoassay for the detection and differentiation of five common carbapenemase families (KPC, OXA-48-like, VIM, IMP, and NDM) in the bacterial colonies, using the NG-Test Carba 5 (NG Biotech, Guipry, France) test. Following the manufacturer’s instructions, we inoculated 100 μL of the cultured suspension in the NG-Test Carba 5 kit using a transfer pipette that was included in the kit. The test was visually examined after 15 min for the presence or absence of the control and test lines, indicating the presence or absence of the respective carbapenemases in the tested specimen

### 2.8. Statistical Analysis

Statistical analysis included descriptive statistics of the tested samples. Categorical variables were compared using the chi square test. Statistical analysis was performed using the Stata 15.0 software (Stata Corp., College Station, TX, USA), and a *p*-value < 0.05 indicated statistical significance. 

## 3. Results

Overall, *K. pneumoniae* strains were detected in 90 (81.8%) samples, whereas *E. coli* strains were detected in 20 (18.2%) samples. PCR analysis confirmed the presence of *K. pneumoniae* strains and *E. coli* strains in positive culture samples (Table 2). Among the 90 samples identified as *K. pneumoniae*-positive, 23 (25.6%) were from chicken, 32 (35.6%) from bovine samples and 35 (38.9%) from pork. Moreover, 23 of 35 (65.7%) chicken samples, 32 of 35 (91.4%) bovine samples, and 35 of 40 (87.5%) pork samples were positive for *K. pneumoniae.* The comparison of contamination rates among the different meat categories revealed that chicken samples were less contaminated than bovine (*p* = 0.008) and pork (*p* = 0.024) samples. However, the rates of *K. pneumoniae* contamination between pork and bovine samples did not differ (*p* = 0.58). 

### 3.1. Phenotypic Analysis 

Based on the in vitro multiplex immunoassay, 54 (60.0%) isolates were positive for the production of at least one carbapenemase. Specifically, 4 (7.4%) *K. pneumoniae* isolates were positive for KPC production, 5 (9.2%) *K. pneumoniae* isolates were positive for VIM production, 32 (59.2%) *K. pneumoniae* isolates were positive for NDM production, 5 (9.2%) *K. pneumoniae* isolates were positive for IMP production, and 17 (31.4%) *K. pneumoniae* isolates were positive for OXA-48 production (Table 3). Moreover, of the 54 isolates with at least one carbapenemase production, there were nine isolates with simultaneous production of two carpapenemases.

### 3.2. Virulence and Resistance Genes

Regarding the virulence genes, 30 (33.3%) *K. pneumoniae* isolates amplified a 759-bp product corresponding to the molecular weight of the *ecpA* gene. The *ecpA* gene was detected in *K. pneumoniae* strains from 14 (46.7%) chicken and 16 (53.3%) bovine specimens. Moreover, the *fimH-1* and *mrkA* genes, encoding type 1 and type 3 fimbrial adhesins, were present in 15 (16.7%) and 65 (72.2%) strains of *K. pneumoniae* isolates, respectively. Among the 15 samples, the *fimH-1* gene was present in 33.3% of the three categories of meat products, as shown in Table 4. Among the 65 isolates, the *mrkA* gene was present in 50 of the chicken and bovine meat products and 15 of the pork meat products. Interestingly, 11.1% (10/90) of the strains possessed the *cap*^+^/*fimH*^+^/*mrkA*^+^ genotype, 0% (0/90) possessed the *ecpA*^+^/*fimH*^+^/*mrkA*^−^ genotype, and 22.2% (20/90) of the strains had the *cap*^+^/*fimH*^−^/*mrkA*^+^ genotype.

Regarding the resistance genes, 65 (72.2%) *K. pneumoniae* isolates were found to carry at least one resistance gene. Of these, *blaNDM*-like was the most predominant gene identified in 40 (61.5%) isolates, followed by *blaOXA*-48-like in 20 (30.7%) isolates, while 5 (7.6%) isolates carried the *blaIMP*-type gene, 5 (7.6%) isolates carried the *blaVIM*-type gene, and 5 (7.6%) isolates carried the *blaKPC*-type gene (Table 3). The frequency of the *blaNDM*-like gene was significantly higher than those of the *blaVIM*-type, *blaKPC*-type, *blaIMP*-type, and *blaOXA-48* genes (*p* < 0.001). Moreover, the frequency of the *blaOXA-48* gene was significantly higher than those of the *blaVIM*-type, *blaKPC*-type and *blaIMP*-type genes (*p* < 0.001). Among the 65 isolates, 10 isolates had two resistance genes simultaneously, 5 had the *blaOXA*-48 and *blaIMP* genes, and 5 had the *blaNDM* and *blaKPC* genes. 

## 4. Discussion

*K. pneumoniae* is not traditionally recognized as a common foodborne pathogen, and most studies investigating foodborne strains are focused on other common pathogens such as *Escherichia coli*, Salmonella and Shigella. Therefore, information regarding the contamination rate of retail food with *K. pneumoniae* strains and their characteristics such as antibiotic resistance and virulence profile that are necessary to evaluate their risk to public health is scarce [20,21,22,23]. In particular, in Greece, there are no reports regarding the prevalence and virulence of *K. pneumoniae* in food supply. It is of great importance to evaluate the genotypic and phenotypic characteristics of these foodborne pathogens and detect their antibiotic resistance since they can cause community and nosocomial outbreaks, similar to other common foodborne pathogens [24]. The evaluation of pathogens in food that harbor antibiotic resistance could allow public health authorities to implement more effective strategies to decrease the contamination rate and mitigate the impact of these pathogens on public health. This is even more prominent in certain areas such as Singapore, where most of the local food supply is imported, and therefore the first stages of production cannot be monitored and controlled for food contamination. Our findings raise concerns regarding the high contamination rate of meat products with *K. pneumoniae* in Greece (81.8%), while also most of the isolated pathogens carried at least one carbapenemase-resistant gene (72.2%), indicating a difficulty in the eradication of these pathogens. Our findings are in line with the reported high rates of antibiotic-resistant *K. pneumoniae* strains in various poultry and meat products, indicating that foodborne *K. pneumoniae* can become a potential pool for antibiotic resistance genes [25].

These pathogens constitute part of the normal human and animal flora, and therefore isolation of *K. pneumoniae* strains in food samples is not necessarily linked with infection transmission. However, their detection in the food chain can be indicative for non-hygienic practices of food preparation and handling, undercooking and suboptimal storage conditions, especially in cases of isolation in ready-to-eat cooked food. Although it has been also suggested that the incidence of liver abscesses due to *K. pneumoniae* worldwide is associated with the geographic distribution of certain virulent *K. pneumoniae* subtypes, the exact route of *K. pneumoniae* infections is not fully elucidated [10,26]. The association between foodborne *K. pneumoniae* and development of infections is not fully investigated, and there is a lack of justification for the incidence of *K. pneumoniae* as a foodborne pathogen. However, since the human gut microflora is closely related to the microbiome in consumed food, and in many cases colonization of the intestine with *K. pneumoniae* preceded infections, foodborne *K. pneumoniae* could be a potential source for the development of *K. pneumoniae* infections in the general population [16]. In line with this, Gorrie et al. found that half of the patients with *K. pneumoniae* infections in the intensive care unit tested positive for gastrointestinal carriage on admission, indicating that gut colonization is a likely entry source for *K. pneumoniae* infections [16].

The first finding emerging from the present study is the high presence of *K. pneumoniae* detected in meat. The prevalence of *K. pneumoniae* that was found in our study is significantly higher than those reported in other studies [20,23]. Guo et al. assessed the frequency of *K. pneumoniae* in food samples in Eastern China and reported a contamination rate of 9.9% (99 positive out of 998 tested samples) [23]. In another study, Hartantyo et al. evaluated the frequency of foodborne *K. pneumoniae* in Singapore, by screening 698 samples of raw and ready-to-eat retail food. The authors detected *K. pneumoniae* in 21% (147 out of 698) of samples tested, which is also significantly smaller than our contamination rate. The different contamination rates between our study and the other studies could be attributed to the heterogeneity in food sources among these studies, since many different categories of food samples such as vegetables, seafood, meat, raw and ready-to-eat food were analyzed those studies. The exact cause for the high frequency of *K. pneumoniae* observed in our study is not easily identifiable, since there are several stages from farmers’ production to consumer markets where infections could have been developed. Thus, further research is needed to clarify this. However, such prevalence of the detected pathogens, and especially in raw food samples, is of utmost importance, as it could be a serious “vector” for the development and the persistence of nosocomial infections. Upon the determination of the primary source responsible for the high contamination rate of meat samples, direct or indirect methods of protection and prevention are necessary. 

Resistance to antibiotics is a multi-faceted and urgent global problem, which needs to be tackled under very different perspectives, involving simultaneously different stakeholders, such as pharmaceutical companies, medical doctors, pharmacies, policy makers, large inter-country health surveillance networks and initiatives, diagnostics companies as well as research institutes [19]. Drug resistance is said to be associated with the spread of transmissible plasmids that also carry virulence genes, and the acquisition of resistant determinants can contribute to the persistence of virulent microorganisms. Carbapenem resistance through carbapenemase production is an emerging threat to public health with a devastating socioecomonic and clinical impact worldwide. Our findings indicate that carbapenem-resistant genes are common among foodborne *K. pneumoniae* strains in Greece. Specifically, we found that 72.2% of the isolated *K. pneumoniae* carried at least one resistance gene for carbapenemase production. The *blaNDM-l* gene was the most predominant antibiotic resistance gene (61.5%), followed by the *blaOXA-48* gene (30.7%). The rate and characteristics of carbapenemase resistance genes has not been evaluated in other studies investigating foodborne *K. pneumoniae*, and therefore we cannot draw any conclusions regarding the cause for the high rate of certain carbapenemase-production genes compared to some others, and we cannot compare our rates with the results of others studies. Guo et al. in his study isolated 99 *K. pneumomiae* strains from different food samples including cooked food, frozen raw food, and fresh raw meet samples and evaluated their resistance to antibiotics [23]. The authors conducted antimicrobial susceptibility testing for several antibiotics including carbapenems and performed PCR analysis for detection of several resistance genes. However, they did not evaluate carbapenemase resistance genes. As opposed to the results of our study, although many of the isolates in Guo et al.’s study showed resistance to other antibiotics such as ampicillin, none of them demonstrated resistance to carbapenems based on antimicrobial susceptibility testing. The authors reported that resistance to antibiotics was significantly higher in isolates from raw meat samples, and therefore the higher rate of resistance to carbapenems in our study may be attributed to the fact that all of our samples were raw meat products. In another study, Hartantyo et al. detected 147 *K. pneumoniae* isolates in raw and ready-to-eat retail food samples and evaluated their susceptibility to antibiotics in 97 of them [25]. Although antibiotic susceptibility testing based on the disk diffusion method was not performed for carbapenems, isolates were screened for the *bla-KPC* gene by PCR analysis. However, the authors did not report the rate of *bla-KPC* positive isolates in their study. Similar to the findings of Guo et al., the authors of this study also did not report on any positive isolate for the bla-KPC gene associated with resistance to carbapenems. 

Another important finding of our study is the high rate of positive pathogens for certain virulence genes such as the *ecpA* gene (33.3%) the *fimH-1* (16.7%) gene and the *mrkA* (72.2%) gene, indicating a high potential for biofilm formation. The *fimH-1* and *mrkA* genes are encoding type 1 and type 3 fimbrial adhesins, while the ecpA gene encodes another pilus protein which is more common among *E. coli* species. These fimbrial structures constitute one of the main mechanisms for adherence to host cells. The multi-fimbrial nature of the isolated pathogens in our study highlights the high virulence potential of these strains, which in combination with the high prevalence of resistance genes can result in difficult-to-eradicate outbreaks. 

There are some limitations of the study that must be addressed. First, the number of analyzed samples is too small to draw safe conclusions regarding the antimicrobial resistance of *K. pneumoniae* in meat products, and therefore large studies are needed to validate our results. Second, the evaluation of the genotype was limited, including only PCR for certain genes, while more advanced molecular techniques such as the whole-genome sequencing method and multi-locus sequence typing (MLST) to decipher the clonal relationship between the food isolates with other published isolated were not performed. Third, we did not document the association between a positive result for carbapenemase production based on the NG-Test Carba 5 and a positive result for carbapenemase resistance genes on PCR analysis, and therefore we cannot evaluate the agreement between these two methods. Last, the phenotype was assessed by only the immunoassay for the detection and differentiation of common carbapenemase enzymes, while further phenotypic characterization including antimicrobial susceptibility testing and the conjugation assay would be valuable.

## 5. Conclusions

In conclusion, we found that there is a high prevalence of *K. pneumoniae* in Greek meat products. Moreover, most of the isolated strains carried carbapenemase resistance genes and virulence genes, indicating a high pathogenic potential of these pathogens and a significant risk to human health. The data reported in this study will aid in our understanding of the potential risk of *K. pneumoniae* in retail food hygiene, food safety, and public health in Greece. However, more extensive research is needed in order to further characterize the frequency and resistance profile of foodborne *K. pneumoniae* strains.

## Figures and Tables

**Figure 1 foods-11-00708-f001:**
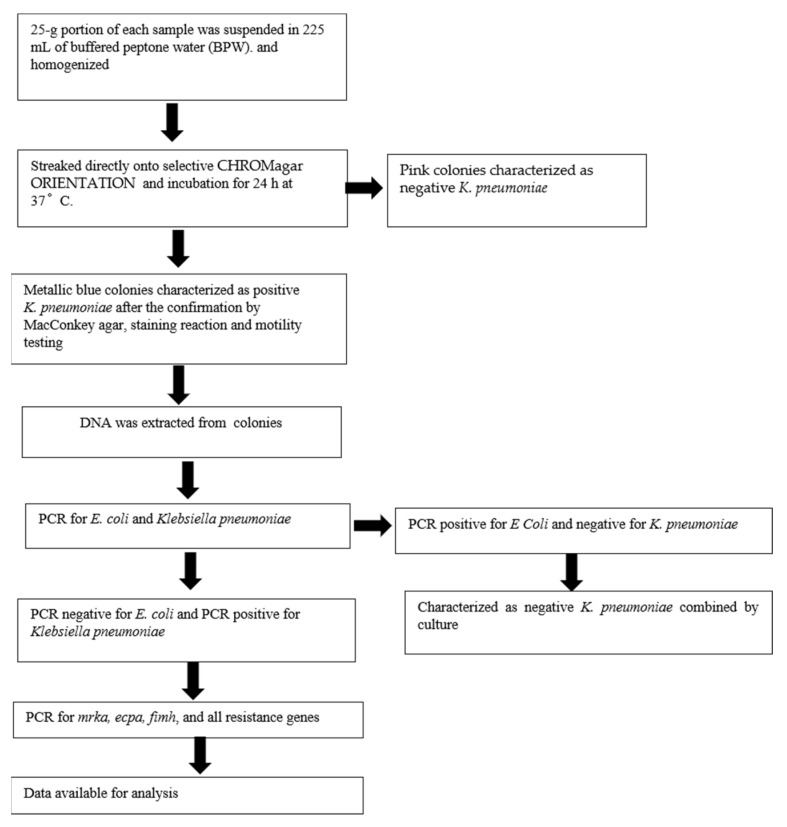
General Workflow. PCR: Polymerase Chain Reaction.

**Table 1 foods-11-00708-t001:** Primers sequences.

Target Genes	Primer Sequence	Ampillicon Size (bp)
*blaIMP-1*	F: 5′-TGA GCA AGT TAT CTG TAT TC-3′	139
	R:5′-TTA GTT GCT TGG TTT TGA TG-3	
*blaOXA-48*	F: 5′-TTG GTG GCA TCG ATT ATC GG-3′	281
	R: 5′-GAG CAC TTC TTT TGT GAT GGC-3′	
*fimH*	F: 5′-CGC CTG GTC CTT TGC CTG CA-3′	817
	R: 5′-CTG CAC GTT GCC GGC GGT AA-3′	
*ecpA*	F: 5′-AAT GGT TCA CCG GGA CAT CAT GTC C-3′	759
	R: 5′-AAG GAT GAA ATA TCG CCG ACA TCC-3′	
*mrkA*	F: 5′-GTT AAC GGC GGC CAG GGC AGC GA-3′ R: 5′-AGG TGA AAC GCG CGC CAT CA-3′	382
*blaVIM*	F: 5′- GAT GGT GTT TGG TCG CAT A-3′	390
	R: 5′-CGA ATG CGC AGC ACC AG-3′	
*blaNDM*	F: 5′-GGT TTG GCG ATC TGG TTT TC-3′	521
	R: 5′- CGG AAT GGC TCA TCA CGA TC-3′	
*blaKPC*	F: 5′-ATG TCA CTG TAT CGC CGT CT-3′	538
	R: 5′-TTT TCA GAG CCT TAC TGC CC-3′	
*E. coli*	F: 5′-TGATTGAAGCAGAAGCCTGC R: 5′-CGCCAATCCACATCTGTGAA	1.350
*Klebsiella pneumoniae*	F 5′-ATTTGAAGAGGTTGCAAACGAT-3′ R 5′-TTCACTCTGAAGTTTTCTTGTGTTC-3′	130

**Table 2 foods-11-00708-t002:** *Klebsiella pneumoniae* and *Escherichia coli* strains in meat samples.

Strains	Overall (* *n* = 110)	Chicken (* *n* = 35)	Bovine (* *n* = 35)	Pork (* *n* = 40)
*Klebsiella pneumoniae*	90 (81.8)	23 (65.7)	32 (91.4)	35 (87.5)
*Escherichia coli*	20 (18.2)	7 (20.0)	3 (8.5)	10 (25.0)

Data are presented as absolute values (percentages). * *n* refers to samples.

**Table 3 foods-11-00708-t003:** Carbapenemase production based on PCR and phenotypic assays.

Carbapenemase	Positive Isolates Based on PCR (* *n* = 65)	Positive Isolates Based on NG-Test Carba 5 (* *n* = 54)
KPC	5 (7.6)	4 (7.4)
OXA-48	20 (30.7)	17 (31.4)
IMP	5 (7.6)	5 (9.2)
VIM	5 (7.6)	5 (9.2)
NDM	40 (61.5)	32 (59.2)

Data are presented as absolute values (percentages). * *n* refers to samples.

**Table 4 foods-11-00708-t004:** Klebsiella pneumoniae strains carrying ecpA, fimH-1 and mrkA genes.

Meat Specimens	*ecpA* (*n* = 30)	*fimH-1* (*n* = 15)	*mrkA* (*n* = 65)
Chicken	14 (46.7)	5 (33.3)	25 (38.5)
Bovine	16 (53.3)	5 (33.3)	25 (38.5)
Pork	0 (0.0)	5 (33.3)	15 (23.1)

Data are presented as absolute values (percentages).

## Data Availability

Data is contained within the article.

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
