# Peer review of "High Prevalence of Klebsiella pneumoniae in Greek Meat Products: Detection of Virulence and Antimicrobial Resistance Genes by Molecular Techniques"

_foods, 2022, doi:10.3390/foods11050708_

Round 1
Reviewer 1 Report
There are some issues with style of writing in this manuscript. For example: avoid use of words dangerous- it sounds unscientific. This study is not at all Innovative, there are numerous studies that are similar to this study.
One of the important drawback of this research was: the number of samples that were collected to draw conclusions about its antimicrobial resistance stains of K. pneumoniae in meat products. To holistically validate at least- 500 samples must be tested.
The methodologies, representation of data and writing are scientifically sound and are well presented. Please check spellings in depth: L296 Antibiotic; and Although both are miss-spelled from which I highly recommend to proof-read the manuscript atleast 9-times. L254 K.p is not italicize. Upon cross-checking the reference 9 and content that was written in L238: "Naturally, these pathogens exist in in soil and their presence is closely associated with farming and agricultural activities since it has been shown that they can result in increased crop yield"
I am surprised both are irrelevant and do not match each other. Hence it can be clearly seen that this manuscript was not revised at all and was not proof read properly. There is lack of justification on the incidence of K.p as a foodborne pathogen and its global incidence (statistics?). Box two in the figure 1 is not clear.
Author Response
Reviewer #1
There are some issues with the style of writing in this manuscript. For example: avoid the use of words dangerous- it sounds unscientific. This study is not at all Innovative, there are numerous studies that are similar to this study.
Author response: We thank the reviewer for his/her comments. The style of the writing has been revised and some words have been replaced according to your suggestion. We agree with the reviewer that there are several studies in the literature regarding the same topic, investigating the virulence and antibiotic resistance of Klebsiella pneumoniae species in food products. However, there is scarce evidence about the prevalence of these pathogens in the Greek market and no reports about their virulence in Greece. In line with the reviewer’s comment, we deleted the word “innovative” from the Conclusion section of the manuscript.
One of the important drawbacks of this research was: the number of samples that were collected to draw conclusions about its antimicrobial resistance strains of K. pneumoniae in meat products. To holistically validate at least- 500 samples must be tested.
Author response: We agree with the reviewer that the small number of samples is a certain limitation of the study. This is now discussed and addressed in the limitation paragraph of the manuscript.
The methodologies, representation of data, and writing are scientifically sound and are well presented. Please check spellings in-depth: L296 Antibiotic; and Although both are miss-spelled from which I highly recommend to proof-read the manuscript at least 9-times. L254 K.p is not italic size.
Author response: We thank the reviewer for his/her kind comment. The manuscript has been screened again for any spelling errors and proofreading has been repeated. The word “antibiotic” has been corrected, and K.p is now in italics.
Upon cross-checking reference 9 and content that was written in L238: "Naturally, these pathogens exist in soil and their presence is closely associated with farming and agricultural activities since it has been shown that they can result in increased crop yield"I am surprised both are irrelevant and do not match each other. Hence it can be clearly seen that this manuscript was not revised at all and was not proofread properly.
Author response: We agree with the reviewer that this phrase does not provide and is relevant to the topic information and has been deleted. The manuscript has been screened, edited, and proofread again to eliminate any inconsistencies between the text and the references, and to correct any spelling errors.
There is a lack of justification on the incidence of K.p as a foodborne pathogen and its global incidence (statistics?).
Author response: The statistics and prevalence of Klebsiella pneumoniae in food samples are discussed in the third paragraph of the Discussion, which was found to be between 9.9% and 21% in other studies, and the issue of the difference between the reported rates in those studies and in our study is further discussed. However, we agree with the reviewer that the association between foodborne K. pneumoniae and the development of infections is not fully investigated. This has been already briefly discussed in the second paragraph of the Discussion section, in which we state that “the exact route of K. pneumoniae infections is not fully elucidated”, and that there are only some indications about this association between foodborne K. pneumoniae and infections, based on the fact that in some studies colonization of the intestine with K. pneumoniae preceded infections. One published study is further discussed, in which the authors found that half of the patients with K.Pneumoniae infections in the Intensive Care Unit were tested positive for gastrointestinal carriage on admission, indicating that gut colonization is a likely entry source for K.Pneumoniae infections [1]. In line with the reviewer’s comment, we have amended the third paragraph of the Discussion section, adding a phrase to highlight the lack of evidence on the incidence of K.p as a foodborne pathogen
- Gorrie CL, Mircea M, Wick RR, Edwards DJ, Thomson NR, Strugnell RA, et al. Gastrointestinal carriage is a major reservoir of Klebsiella pneumoniae infection in intensive care patients. Clin Infect Dis 2017; 65(2):208-215.
Box two in figure 1 is not clear.
Author response: The figure has been amended and now all boxes are clear. Thank you.
Reviewer 2 Report
This study investigated and found a high presence of K. pneumoniae in Greek meat samples. K. pneumoniae isolates producing carbapenemases or carrying carbapanemase resistance genes and virulence genes were tested.
- Several virulence genes carried by pneumoniae isolates were tested to indicate the pathogenic potential to human health. A brief introduction or discussion should be given to explain these virulence genes.
- “2.9 general workflow“, only a chart there.
- As shown in table 3, how about the correspondence between carbapenemase-positive isolates and resistance genes positive isolates?
- There are many typos. prevelence (Line 20), Coli (E. coli), Bla (table 1. bla), etc.
Author Response
Reviewer #2
Several virulence genes carried by pneumoniae isolates were tested to indicate the pathogenic potential to human health. A brief introduction or discussion should be given to explain these virulence genes.
Author response: We agree with the reviewer that information regarding these virulence genes would be helpful for the readers. A brief description of these genes is now provided in the Discussion section.
“2.9 general workflows “, only a chart there.
Author response: This figure has been amended.
As shown in table 3, how about the correspondence between carbapenemase-positive isolates and resistance genes positive isolates?
Author response: We agree with the reviewer that the correspondence between carbapenemase-positive isolates based on NG-Test Carba 5 and resistance genes positive isolates would be valuable for the readers. Unfortunately, we did not document the association between the positive result for carbapenemase production based on NG-Test Carba 5 and the positive result for carbapenemase resistance genes on PCR analysis (we were not recording whether the specific strain that was positive/negative on the NG-Test Carba 5 was positive/negative on the PCR analysis). This is now addressed in the limitation paragraph of the Discussion section in which we report: “Third, we did not document the association between a positive result for carbapenemase production based on the NG-Test Carba 5 and a positive result for carbapenemase resistance genes on PCR analysis, therefore we cannot evaluate the agreement between these two methods”.
There are many typos. prevalence (Line 20), Coli (E. coli), Bla (table 1. bla), etc.
Author response: We thank the reviewer for his/her comment. The manuscript has been screened again for any spelling errors and proofreading has been repeated. The above typos have been corrected.
Reviewer 3 Report
The manuscript by Theocharidi et al. describe the study of the incidence of Klebsiella pneumonia in Greek meat product and their association with antimicrobial resistance by the phenotypic and molecular analysis. The work showed intense analysis, and the results presented are relevant for the field. Critical analysis to the study was performed by the authors showing the importance of the work, minor revisions needed.
Few comments raised below would help the authors to revise the manuscript and clarify some of the information.
Line 46: community? Do the authors mean ‘comorbidities’
Figur 1. The figure need to be uploaded as image. Some of the text is hidden
Line 111:PBW 1%? In the Figure 1, BPW is referred.
Line 129 and 143: only want to confirm that the concentration of the primers is correct. 1mM seems a really high concentration. Also you should rewrite the “1mM each of gene specific primers” part to a more clear expression.
Line 166: This section need to have some info written, as the figure don’t explain anything. The Scheme also need to be labelled as Figure 2 and referenced in the text.
Table 2 and 3: authors should indicate what mean the “n”.
The results obtained in table 3 need to be further explained and discussed. Mostly regarding the incidence of each gene and why some of them have higher positive results that others…it’s this comparable with reported results in other studies.
Author Response
Reviewer #3
The manuscript by Theocharidi et al. describes the study of the incidence of Klebsiella pneumonia in Greek meat product and their association with antimicrobial resistance by the phenotypic and molecular analysis. The work showed intense analysis, and the results presented are relevant for the field. Critical analysis of the study was performed by the authors showing the importance of the work, minor revisions needed.
Author response: We thank the reviewer for his/her kind comments
Line 46: community? Do the authors mean ‘comorbidities’
Author response: We thank the reviewer for his/her comment. We meant to say community-acquired infections. This has been corrected in the revised manuscript.
Figure 1. The figure needs to be uploaded as an image. Some of the text is hidden
Author response: This figure has been amended and is now uploaded as an image.
Line 111:PBW 1%? In Figure 1, BPW is referred to.
Author response: We thank the reviewer for noticing the error. PBW in the manuscript has been corrected to BPW.
Line 129 and 143: only want to confirm that the concentration of the primers is correct. 1mM seems a really high concentration. Also, you should rewrite the “1mM each of gene-specific primers” part to a more clear expression.
Author response: We thank the reviewer for noticing the error. 1 mM concentration has been corrected to 0.2μM in the revised manuscript.
Line 166: This section needs to have some info written, as the figure doesn’t explain anything. The Scheme also needs to be labeled as Figure 2 and referenced in the text.
Author response: The subtitle “General workflow” in this line was a typo error since it was repeated by mistake. The subsection entitled “General workflow” is at the beginning of the Methods section with a respective text and figure (Figure 1), describing the study workflow. The misplaced subtitle “General workflow” has been deleted from the manuscript. The scheme has been also deleted from the revised manuscript since it does not provide any useful information to the readers.
Table 2 and 3: authors should indicate what means “n”.
Author response: The letter “n” is now explained as a footnote in tables 2 and 3.
The results obtained in table 3 need to be further explained and discussed. Mostly regarding the incidence of each gene and why some of them have higher positive results than others…it’s this comparable with reported results in other studies.
Author response: We agree with the reviewer that there is high heterogeneity in the rate of carbapenemase production genes among the isolates and that it would be valuable for the readers to compare these rates with the reported results of other published studies. However, the rate of carbapenemase resistance genes has not been evaluated in other studies investigating foodborne K.pneumoniae, therefore we cannot draw any conclusions regarding the cause for the high rate of certain carbapenemase-production genes in our study, and we cannot compare our rates with the results of others studies. Although resistance to carbapenems has been evaluated in some other studies investigating foodborne K. pneumoniae, whether a PCR analysis was not performed in these studies or a PCR analysis was performed but the rates of these genes were not reported [1,2]. According to the reviewer’s suggestion, this issue is now further discussed in the fourth paragraph of the Discussion section.
- Guo Y, Zhou H, Qin L et al. Frequency, Antimicrobial Resistance and Genetic Diversity of Klebsiella pneumoniae in Food Samples. PLoS One. 2016;11(4):e0153561.
- Sri Harminda Pham Hartantyo; Manling Chau; Tse Hsien Koh; Min Yap; Tseng Yi; Delphine yan Hong Cao Ramona Alikiiteaga Gutiérrez; Lee ching Ng. 2020. Foodborne Klebsiella pneumoniae: Virulence Potential, Antibiotic Resistance, and Risks to Food SafetyJ Food Prot, 83 (7): 1096–1103.
Round 2
Reviewer 1 Report
Authors have not completed their review correctly. Words such as 'Innovative' was not deleted in the revised manuscript.
Author Response
Authors have not completed their review correctly. Words such as 'Innovative' was not deleted in the revised manuscript
Author response: We thank the reviewer for his/her comments. As the reviewer noticed, although the word “innovative” was deleted from the Conclusions section, it was not deleted from the Introduction section. We have now deleted the word “innovative” from the Introduction section as well, therefore this word does not appear anywhere in the manuscript.